# Semantic similarity prediction is better than other semantic similarity measures

**Steffen Herbold**                                                    *steffen.herbold@uni-passau.de*
*Faculty of Computer Science and Mathematics*
*University of Passau*
*Passau, Germany*

**Reviewed on OpenReview:** *https://openreview.net/forum?id=bfsNmgN5je*

## Abstract

Semantic similarity between natural language texts is typically measured either by looking at the overlap between subsequences (e.g., BLEU) or by using embeddings (e.g., BERTScore, S-BERT). Within this paper, we argue that when we are only interested in measuring the semantic similarity, it is better to directly predict the similarity using a fine-tuned model for such a task. Using a fine-tuned model for the Semantic Textual Similarity Benchmark tasks (STS-B) from the GLUE benchmark, we define the STSScore approach and show that the resulting similarity is better aligned with our expectations on a robust semantic similarity measure than other approaches.

## 1 Introduction

When we are considering research questions related to the quality of the output of generative models for natural language (e.g., Large Language Models/LLMs like GPT(-2) by Radford et al. (2019) and all the other currently very successful models that followed), we need to determine if the generated text is a valid output for the desired task. For example, when we want to use such a model for translation from English to German, we need to determine if the resulting German sentence has the same meaning as the English sentence. Since there are typically many possible valid outputs for such tasks, we cannot easily specify a ground truth for benchmarks. While an expert would be able to determine if the output is correct, automating this task with a computer is not easy, as this would effectively require solving the translation problem first. However, since scaling such evaluations to thousands or even millions of examples is not possible without automation, researchers created different methods to assess the quality of output using the idea of *semantic similarity*.

The general idea is quite simple: while we cannot define the ground truth, we can measure the semantic meaning to one or more correct outputs. Simple metrics such as the Word Error Rate (WER) estimate this through the consideration of the differences between words, similar to the Levenshtein distance. Others, like BiLingual Evaluation Understudy (BLEU, Papineni et al. (2002)) or Recall-Oriented-Understudy for Gisting Evaluation (ROGUE, Lin (2004)) are rather looking at the overlap between subsequences of texts, assuming that a larger overlap means a better semantic similarity. Furthermore, some methods (e.g., BLEU) go beyond the comparison of single solutions to a sample output and instead allow the comparison between the corpora of generated and sample solutions. For individual words without considering them in their context, word embeddings like word2vec (Mikolov et al., 2013) in combination with cosine similarity are a generally accepted way to estimate the semantic similarity between words, even though this approach has problems with ambiguities, e.g., caused by polysemes (Del Tredici & Bel, 2015). Similar methods were extended to whole sentences (Sharma et al., 2017), but they did not achieve the same level of performance.

The transformer architecture enabled the context-sensitive calculation of embeddings (Vaswani et al., 2017). Naturally, this was adopted for the calculation of the similarity. Two methods based on this are currently primarily used. The first is BERTScore by Zhang et al. (2020), which is based on an optimal matching of the pair-wise similarities of words within a contextual BERT embedding and has been used by thousands of

publications since its publication in 2020. The second is Sentence-BERT (S-BERT) by Reimers & Gurevych (2019), who pool the embeddings of the tokens into an embedding for sentences. The cosine similarity can then be computed between these sentence embeddings, the same as between words with word2vec or with earlier, less powerful, sentence embeddings (e.g. Sharma et al., 2017).

This success notwithstanding, we want to argue that this approach should not be further pursued in favor of a simpler solution to estimate the semantic similarity, i.e., directly predicting the semantic similarity with a regression model. We note that this idea is not new and was, e.g., also used to define approaches like BEER (Stanojević & Sima'an, 2014) or RUSE (Shimanaka et al., 2018). However, these models are from the pre-transformer era of natural language processing. As can be seen in large benchmarks like GLUE (Wang et al., 2018) and SuperGLUE (Wang et al., 2019), models based on the transformer architecture (Vaswani et al., 2017) provide a much better performance. Therefore, we formulate the following hypothesis for our research:

**Hypothesis:** Modern language models with an encoder-only transformer architecture similar to BERT (Devlin et al., 2019) that are fine-tuned as regression models for the similarity between sentence pairs are also capable of robustly measuring the semantic similarity beyond their training data and are better measures for the semantic similarity than embedding-based and n-gram approaches.

We derive this hypothesis from the assumption that if such a regression model fulfills its task, employing it as a semantic similarity measure would be the natural use case beyond just benchmarking model capabilities. Within this paper, we present the results of a confirmatory study that demonstrates that downloading a fine-tuned RoBERTa model (Liu et al., 2019) for the STS-B task (Cer et al., 2017) from the GLUE benchmark from Huggingface and using this model to predict the similarity of sentences fulfills our expectations on a robust similarity measure better than the other models we consider. We refer to this approach as STSScorer. To demonstrate this empirically, we compute the similarity score for similarity related GLUE tasks and show that while the predictions with the STSScorer are not perfect, the distribution of the predicted scores is closer to what we would expect given the task description, than for the other measures.

## 2 Method

Within this section, we describe the research method we used to evaluate our hypothesis. We first describe the STSScorer model for the prediction of the similarity in Section 2.1. Then, we proceed to describe our analysis approach in Section 2.2, including the expectations we have regarding our hypothesis and the tools we used in Section 2.6.

### 2.1 STSScorer

Listing 1 describes our approach: we download a fine-tuned model from Huggingface for the STS-B task (Held, 2022), which was based on RoBERTa (Liu et al., 2019). The STS-B task contains sentence pairs from news, image captions and Web forums. Each sentence pair received a score between zero and five. These scores were computed as the average of the semantic similarity rating conducted by three humans such that five means that the raters believe the sentences mean exactly the same and zero means that the sentences are completely unrelated to each other. We simply use a regression model trained for this task and divide the results by five to scale them to the interval $[0, 1]$. The regression model is trained with Mean Squared Error (MSE) loss, which is defined as $MSE(y, y^*) = (y - y^*)^2$, where $y$ is the predicted label and $y^*$ the expected label. Hereafter, we refer to this model as STSScorer.

### 2.2 Analysis approach

Our analysis approach is similar to the method used by Zhang et al. (2020) for the evaluation of the robustness of similarity measures also by Reimers & Gurevych (2019) for the evaluation of S-BERT: we utilize data labeled data for which we have an expectation of what to observe when measuring the semantic similarity.

Listing 1: A simple class to define a fully functional semantic similarity scorer based on a pre-trained model for the STS-B tasks.

```python
import transformers

class STSScorer:
    def __init__(self):
        model_name = 'WillHeld/roberta-base-stsb'
        self._sts_tokenizer = transformers.AutoTokenizer.from_pretrained(
            ↪ model_name)
        self._sts_model = transformers.AutoModelForSequenceClassification.
            ↪ from_pretrained(model_name)
        self._sts_model.eval()

    def score(self, sentence1, sentence2):
        sts_tokenizer_output = self._sts_tokenizer(sentence1, sentence2, padding=
            ↪ True, truncation=True, return_tensors="pt")
        sts_model_output = self._sts_model(**sts_tokenizer_output)
        # logits contain regression values
        # need to divide by five due to scoring approach of STS-B between 0 and 5
        return sts_model_output['logits'].item()/5
```

There are four such data sets within the GLUE benchmark:

- the Semantic Textual Similarity Benchmark (STS-B) data we already discussed above;

- the Microsoft Research Paraphrase Corpus (MRPC, Dolan & Brockett (2005)) data, where the task is to determine if two sentences are paraphrases;

- the Quora Question Pairs (QQP, Iyer et al. (2017)) data, where the task is to determine if two questions are duplicates; and

- the Chinese to English translations from the WMT22 metrics challenge (WMT22-ZH-EN, Freitag et al. (2022)), where the translation quality is labeled using the MQM schema Burchardt (2013).

For STS-B, MRPC, and WMT22, we use the test data. Since the labels for QQP's test data are not shared, we use the training data instead. To the best of our knowledge, this data was not seen during the training of the STSScorer and S-BERT models of the models we use, as Quora was not part of the pre-training corpus of RoBERTa, which mitigates the associated risks regarding data contamination. However, the model underlying S-BERT was fine-tuned using contrastive learning on a corpus of one billion sentences (Reimers & Gurevych, 2019), which contained about 100,000 instances from the QQP data, i.e., about a quarter. Thus, S-BERT might have an advantage on this data.

On each of these data sets, we compute the similarity between all pairs of sentences with BLEU, BERTScore, S-BERT,[1] and STSScore. All methods we consider compute scores between zero (not similar at all) and one (same semantic meaning), which simplifies the direct comparison. This narrower view of few models allows us to consider the results more in-depth. Specifically, we can go beyond the plain reporting of numbers and instead look directly at the distributions of the similarity measures for different data sets. Due to the confirmatory design of our study, we formulate concrete expectations on the results, given the properties of each data set. How well the similarity measures fulfill these expectations will be later used to evaluate our hypothesis. Based on the strong performance of BERT-based models on the STS-B task in the GLUE benchmark, we predict, based on our hypothesis that STSScorer should have the best alignment with our expectations for all data sets.

---

[1]From now on, we simply refer to calculating the cosine similarity between embeddings with S-BERT as S-BERT for brevity.

We note that while we could have added more approaches to this comparison, e.g., ROUGE (Lin, 2004), METEOR (Lavie & Agarwal, 2007), RUSE (Shimanaka et al., 2018), and BEER (Stanojević & Sima'an, 2014), these models were all already compared to BERTScore, which was determined to provide a better measure for the similarity (Zhang et al., 2020). Further, we refer to a general overview of such metrics to the work by Zhang et al. (2020). Thus, instead of providing a broad comparison with many models, we rather compare our approach to the embedding-based approaches S-BERT and BERTScore which are currently used as de-facto state-of-the-art by most publications and the still very popular BLEU method that uses an n-gram matching approach.

Additional analyses regarding the impact of the sequence lengths, the use of different methods as an ensemble, as well as using a different model within the STSScorer can be found in the appendix.

## 2.3 STS-B data

While the comparison on the STS-B data may seem unfair, because the STSScorer was specifically trained for that model, the analysis of the behavior of the different scorers on this model still gives us interesting insights: for any semantic similarity measure, the distribution of the scores should be directly related to the label of STS-B,[2] which is a human judgment of the semantic similarity. Consequently, when we plot the label on the x-axis versus a semantic similarity score on the y-axis, we would ideally observe a strong linear correlation. Visually, we would observe this by the data being close to the diagonal. A less ideal, but still good, result would be that the trend is monotonously increasing, indicating a rank correlation, which would mean that while the magnitudes of the similarity measure are not aligned with the human judgments from STS-B, at least the order of values is. Any other trend would mean that the similarity measure is not aligned with the human judgments of STS-B. In addition to this visualization, we also measure the linear correlation between the scores and the labels with Pearson's $r$ and the rank correlation with Spearman's $\rho$, as is common for the STS-B task within the GLUE benchmark (Wang et al., 2018) and was also used by Reimers & Gurevych (2019) for the evaluation of S-BERT.

Because the STSScorer was fine-tuned on the STS-B data, we only utilize the test data. Nevertheless, because this is exactly the same context as during the training of STSScorers models (same data curators, same humans creating the judgments) this model has a huge advantage over the other approaches. Due to this, we expect that STSScorer is well aligned with human judgments and we observe the linear trend described above. If this fails, this would directly contradict our hypothesis, as this would not even work within-context. The BLEU, BERTScore, and S-BERT models were created independently of the STS-B data, but given their purpose to estimate the semantic similarity, they should still be able to fulfill the desired properties. If this is not the case, this would rather be an indication that these models are not measuring the semantic similarity – at least not according to the human judgments from the STS-B data.

## 2.4 MRPC and QQP data

The MRPC and QQP data are similar: both provide binary classification problems. With MRPC, the problem is paraphrasing. With QQP, the problem is duplicate questions, which can also be viewed as a type of paraphrasing, i.e., the paraphrasing of questions. Paraphrasing is directly related to semantic similarity, as paraphrased sentences should be semantically equal. Thus, similarity measures should yield high values for paraphrases and duplicate questions, ideally close to one.

For the negative examples of MRPC, a look at the data helps to guide our expectations. When considering the MRPC data, we observe that the negative samples are all sentences on the same topic, with different meaning. As an example, consider the first negative example from the training data:

**Sentence 1:** *Yucaipa owned Dominick 's before selling the chain to Safeway in 1998 for $ 2.5 billion*

**Sentence 2:** *Yucaipa bought Dominick 's in 1995 for $ 693 million and sold it to Safeway for $ 1.8 billion in 1998 .*

---

[2]While STS-B is a regression task and and it would be better to speak of a dependent variable here, we rather speak of labels all the time to be consistent with the subsequent classification tasks.

Both sentences are related to the ownership of *Dominick's* by *Yucaipa* and the sale to *Safeway*, but consider different aspects regarding the time and different amounts of money for the payment. Thus, we observe semantic relationship, but not a paraphrasing. While we have not read through all the negative examples from the MRPC data, we also did not find any examples where both sentences were completely unrelated. Consequently, we expect values for the semantic similarity of the negative examples to be significantly larger than zero, but also smaller than the positive examples of actual paraphrases with a notable gap.

For the negative examples of QQP, the expectation is less clear. For most pairs, we observe that both questions are somewhat related, i.e., different questions regarding the same topic. As an example, consider this negative example from the training data:

**Sentence 1:** *What causes stool color to change to yellow?*

**Sentence 2:** *What can cause stool to come out as little balls?*

Both questions are about *stool*, but different aspects of stool. This kind of difference is similar to what we have within the MRPC data. However, we also observed examples, where the questions are completely unrelated. Consider the following instance from the training data of QQP:

**Sentence 1:** *How not to feel guilty since I am Muslim and I'm conscious we won't have sex together?*

**Sentence 2:** *I don't beleive I am bulimic, but I force throw up atleast once a day after I eat something and feel guilty. Should I tell somebody, and if so who?*

While both questions are broadly related to the concept of *guilt*, the rest is completely unrelated and we would expect a very low semantic similarity. Consequently, while our expectation for the semantic similarity measures for the majority of the negative samples is similar to that of MRPC (i.e., significantly larger than zero, but smaller than for the positive examples), we also expect to observe a strong tail in the distribution with lower similarities.

For both data sets, we visualize the distribution of the similarities per class. Additionally, we report the central tendency (arithmetic mean) and variability (standard deviation) per class in the data. Finally, we consider how the similarity metrics would fare as classifiers based on the Area Under the Curve (AUC).

## 2.5 WMT22-ZH-EN

With the WMT22-ZH-EN data, we consider how the semantic similarity between a reference solution and alternative translations relates to manually labeled translation quality. Semantic similarity between the source and a translation is an important aspect when we want to consider the quality of the generated translation. However, this is only one aspect of the Multidimensional Quality Metrics (MQM) framework for the analytic evaluation of translation quality (Burchardt, 2013). This framework uses multiple criteria such as the accuracy, fluency, terminology, style, and locale conventions. While the semantic similarity is directly related to the accuracy criterion, the others are rather regarding how a translations worded beyond its meaning – something we typically want to ignore when measuring the semantic similarity. Nevertheless, embedding-based approaches for computing the semantic similarity were very successful in past challenges on ranking machine translation systems (Freitag et al., 2022). Based on this, we expect that we can observe correlations between the semantic similarity and the translation quality, but we also expect that the observed correlations are weaker than those for the other data sets, because aspects like fluency, terminology, style, and locale conventions, are not considered by the semantic similarity.

We report the correlations in the same manner as for STS-B, i.e., we visualize the relationship between the human judgments and the semantic similarity and report Pearson's $r$ and Spearman's $\rho$.

## 2.6 Tools used

We used the Huggingface transformer library to implement STSScore and the Huggingface evaluation library for BLEU. The RoBERTa model we used for STSScore (Held, 2022) was trained with a learning rate of

| | STS-B | | MRPC | | QQP | | WMT22-ZH-EN | |
|---|---|---|---|---|---|---|---|---|
| | *r* | *ρ* | *neg* | *pos* | *neg* | *pos* | *r* | *ρ* |
| BLEU | 0.34 | 0.32 | 0.26 (0.19) | 0.39 (0.20) | 0.11 (0.23) | 0.18 (0.25) | 0.14 | 0.13 |
| BERTScore | 0.53 | 0.53 | 0.53 (0.15) | 0.68 (0.13) | 0.44 (0.26) | 0.67 (0.17) | 0.32 | 0.37 |
| S-BERT | 0.83 | 0.82 | 0.71 (0.15) | 0.83 (0.12) | 0.56 (0.23) | 0.86 (0.10) | 0.19 | 0.24 |
| STSScore | 0.90 | 0.89 | 0.61 (0.18) | 0.84 (0.13) | 0.44 (0.24) | 0.76 (0.18) | 0.27 | 0.34 |

Table 1: Summary statistics of the results. Pearson's $r$ and Spearman's $ρ$ between the labels and similarities for STS-B and WMT22-ZH-EN. Mean values with standard deviation in brackets for both classes of the MRPC and QQP data. We use *neg/pos* to indicate the classes such that *pos* is the semantically equal class. All values are rounded to the second digit.

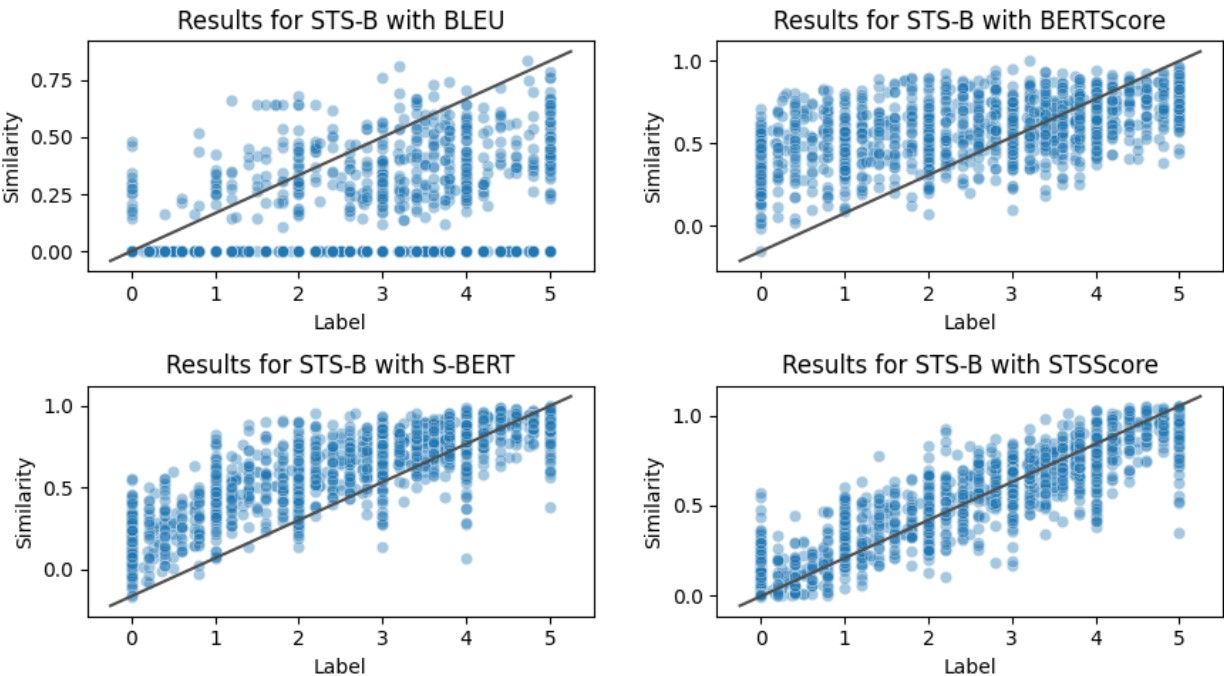

Figure 1: Evaluation of similarity measures on the test data of STS-B. Ideally, the similarity correlates linearly with the labels, i.e., the scores are close to the black line.

$2 \cdot 10^{-5}$, a linear scheduler with a warmup ratio of 0.06 for 10 epochs using an Adam optimizer with $\beta_1 = 0.9, \beta_2 = 0.999$, and $\epsilon = 10^{-8}$ using MSE loss. For S-BERT, we used the `all-MiniLM-L6-v2` which was tuned for high-quality sentence embeddings using a constrative learning model (Reimers, 2021; Reimers & Gurevych, 2019) and the python package provided by Reimers & Gurevych (2019). For BERTScore we used the default RoBERTa model for the English language and the python package provided by Zhang et al. (2020). We used Seaborn for all visualizations and Pandas for the computation of the correlation coefficients, mean values, and standard deviations. All implementations we created for this work are publicly available online: `https://github.com/aieng-lab/stsscore`

## 3   Results

Figure 1 shows the results on the STS-B test data. The STSScore has the expected strong linear correlation with the labels. However, this is not surprising since the underlying model was fine-tuned for this task and the strong performance was already reported through the GLUE benchmark. Still, this confirms that the semantic similarity prediction fulfills the expectations that we have on a semantic similarity measure.

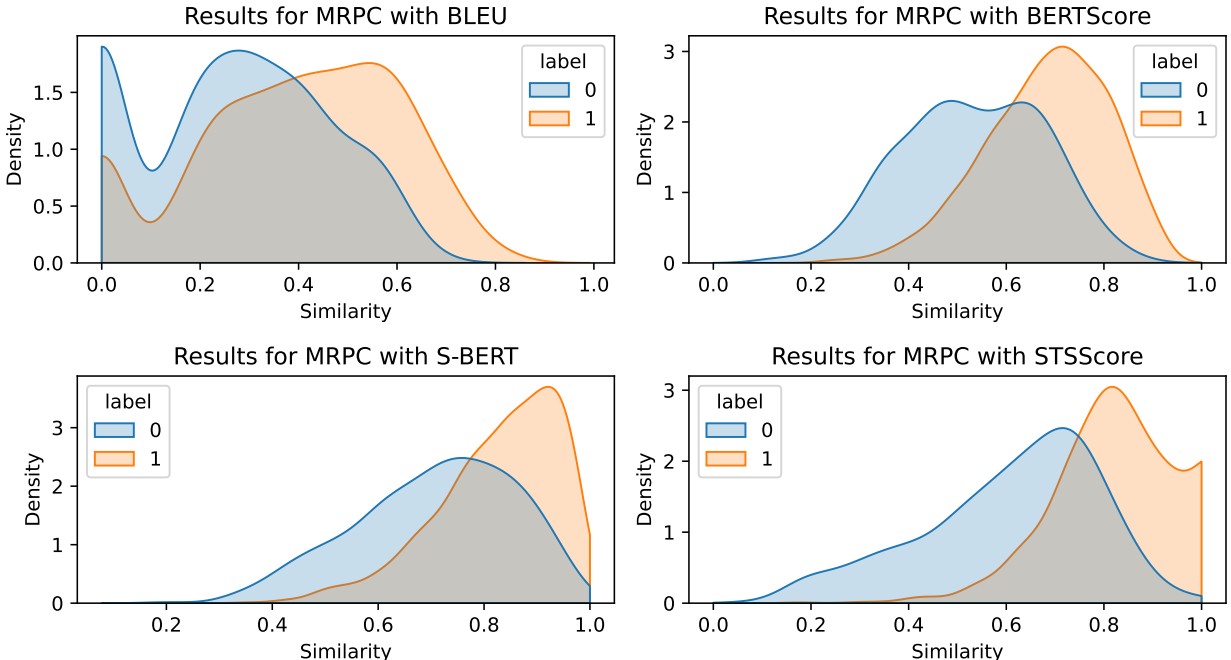

Figure 2: Evaluation of similarity measures on the test data of MRPC. Ideally, the positive class (1) has scores close to one and the negative class (0) has smaller values, but not close to zero.

S-BERT also seems to have the desired linear correlation, but with a general tendency to overestimate the similarity as most values are above the diagonal. The same cannot be said about BLEU or BERTScore. Both are not aligned at all with the expectations from the STS-B task. The values of BERTScore rather seem fairly randomly distributed, the values of BLEU are often exactly zero and otherwise often a lot lower than expected. An optimistic reading of the BERTScore results detects a weak upward slope in the similarity scores that would be expected. The correlation coefficients depicted in Table 1 match the results of our visual analysis: STSScore is strongly correlated ($r = 0.90$, $\rho = 0.89$), followed by S-BERT that is also strongly correlated, but generally a bit weaker than STSScore ($r = 0.83$, $\rho = 0.82$). BERTScore has only a moderate correlation ($r = 0.53$, $\rho = 0.53$) and the correlation of BLEU is weak ($r = 0.34$, $\rho = 0.32$).

Figure 2 shows the results for the MRPC data, Table 1 shows the statistical markers. STSScore yields the expected results: the paraphrasings have a higher similarity than the negative examples, with typically high scores (mean=0.84). However, the density plot shows that the scores are not always close to one, though only few scores are below 0.6. We also observe that the non-paraphrasings are almost always detected semantically somewhat similar (mean 0.61) with a high variability that covers nearly the complete range. However, we note that the density drops to almost zero at very high values ($>0.9$) and very low values ($<0.1$). This is in-line with our expectations: the similarity measure typically does not indicate unwarranted equality and it picks up the relationships within the data not dropping to zero. S-BERT is generally similar with high scores for the paraphrasings (mean=0.83). The distribution looks a bit different from STSScore: while STSScore has a peak at around 0.82 and a second peak at exactly one, S-BERT has only a single peak at about 0.92, but drops sharply after this peak. The non-paraphrasings have a higher semantic similarity for STSScore (mean=0.71), which aligns with the tendency of S-BERT to overestimate the similarity that we also observed with the STS-B data.

The results of BERTScore exhibit a lot of the expected properties, i.e., a larger mean value for the paraphrases and the similarity for the negative examples covers the whole range, except for the very high and very low values. However, we note that there are only few cases with a similarity close to one for the paraphrases, even though this is actually the expected value. Moreover, the peak of the distribution is also a bit lower than for STSScore and the tail towards lower values for paraphrases is also stronger. As a result, the mean

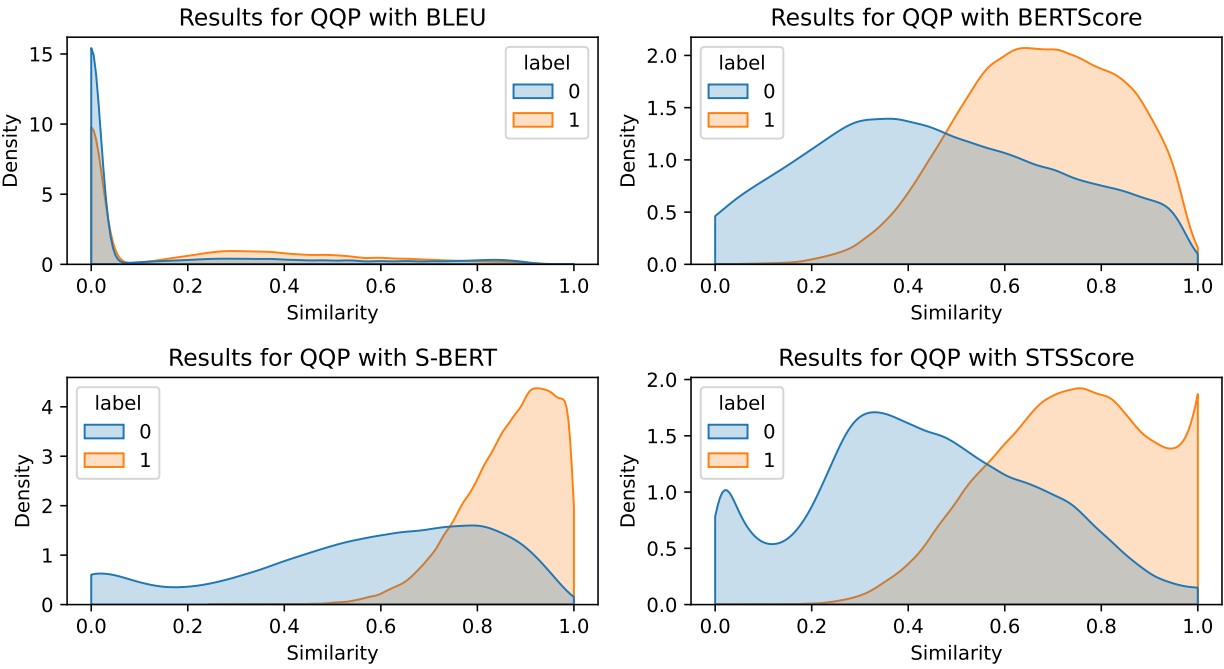

Figure 3: Evaluation of similarity measures on the training data of QQP. Ideally, the positive class (1) has scores close to one and the negative class (0) has smaller values, with only a small fraction being close to zero.

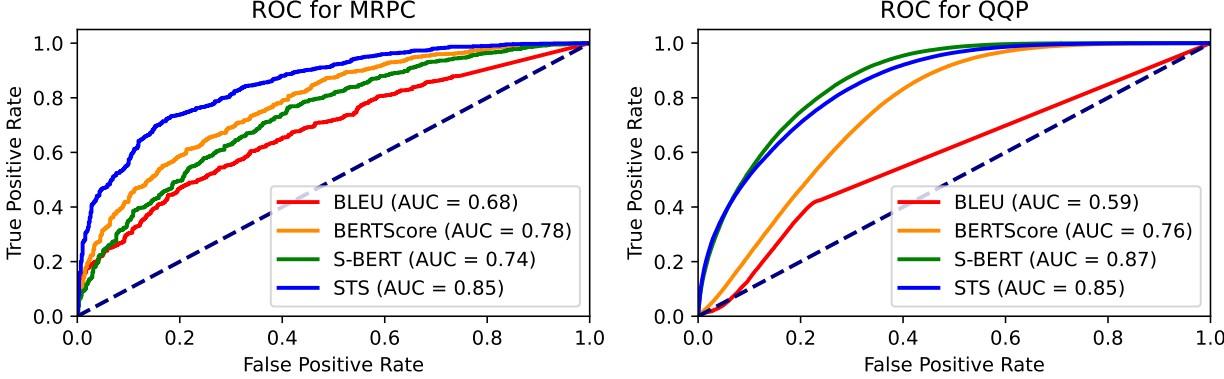

Figure 4: Receiver Operator Characteristics (ROC) and AUC measurements for using the semantic similarity metrics as classifiers for the MRPC and QQP data. A larger area is better.

value for the paraphrases of BERTScore is only 0.68 for the positive examples. Moreover, these downward shifts in the distributions also lead to a larger overlap between the distributions for the paraphrases and the negative examples, i.e., there is a stronger random element in observing large scores with BERTScore than with STSScore.

For BLEU, the results are not fully aligned with our expectations for the MRPC data. BLEU displays the same tendency for often reporting similarities of exactly zero for both classes that we also observed for STS-B. Similarly, the values for both classes are fairly low, with only a mean similarity for the paraphrases of 0.39. However, the visual analysis shows that this mean value is somewhat skewed by the many values that are exactly zero, as the peak for the non-zero similarities is rather around 0.6. Moreover, both the distribution as well the lower mean value (0.26) indicate that the negative examples receive lower similarity scores, as expected. Still, based on the visual analysis, the distributions of the positive and negative samples strongly overlap, meaning that while the score trends in the expected direction at scale, it is not suited for individual results or calibrated as would be expected for this data.

For QQP, the results for all three models are comparable with respect to their alignment with our expectations to the MRPC: STSScore matches our expectations very well. We observe both a lot of very high values, and overall a rather high semantic similarity for the duplicate questions. The mean is a bit lower than for MRPC. However, this seems to be a property of analyzing questions in comparison to regular sentences, as we observe such a downward shift across all classes and similarity measures. We also observe the expected trend towards very low values. S-BERT is a bit of a mixed bag for QQP. On the one hand, the results for the duplicate questions indicate a better measurement of the similarity than for STSScorer. On the other hand, the negative examples also receive higher similarity scores of the same amount and also have their peak at a very high similarity of 0.8, though the distribution is spread out such that it is almost uniform between about 0.5 and about 0.8. When we consider these results in the context of the other data sets, this can be explained by the general property of S-BERT to produce higher values for the measurement of the similarity. We note that S-BERT has seen some of the data from QQP during the pre-training, which may be the reason for the very high similarity of the duplicates. This advantage notwithstanding, it seems that STSScore and S-BERT are comparable on the QQP data, with a stronger separation observed with STSScore, but higher similarities for duplicates with S-BERT.

BERTScore is again exhibiting a lot of the expected properties, but again fails to achieve very large values and has an overall lower similarity for the duplicate questions than STS-B. Same as above, this leads to a larger overlap between the distributions of the classes. The tendency to produce values of exactly zero is strongest for the QQP data. In general, one can say that BLEU failed for this data set: while there is still some difference in the mean value, most similarity scores of BLEU are exactly zero for both classes.

Figure 4 provides another perspective on the results for the MRPC and QQP data, i.e., the performance measured through the AUC when using a threshold to directly classify the data given the similarity measures. This perspective augments the distributions we have analyzed above, as the overlap between the distributions is the cause for misclassifications for different thresholds. Consequently, a smaller overlap that allows for a better separation between the paraphrases, respectively duplicate questions and leads to a larger AUC. This ROC analysis supports the above conclusions regarding the differences: STSScorer is clearly the best for MRPC, followed by S-BERT, BERTScore, and BLEU, all with notable gaps. For the QQP data, STSScorer and S-BERT perform similar. As we observed before, the other models both perform notably worse.

Figure 5 shows the correlation between semantic similarity measures and the manually determined MQM judgments on the WMT22-ZH-EN data. The embeddings based approaches all correlate somewhat with the MQM judgments, but there is a huge spread within the similarity values for the same MQM judgments. BERTScore achieves the best correlation and has the lowest spread within the visualization. STSScore has a slightly lower correlation, which also shows through a larger spread of similarity values compared to BERTScore. S-BERT performs worse than the other approaches with a notable gap. A potential reason for the stronger performance of BERTScore is that the matching that is performed based on pair-wise similarities of words may align well with the MQM criteria beyond accuracy, as this allows to measure differences on the word level, while STSScore and S-BERT consider the full text at once. We do not have a clear explanation why S-BERT performs so much worse than the other embedding-based approaches. All

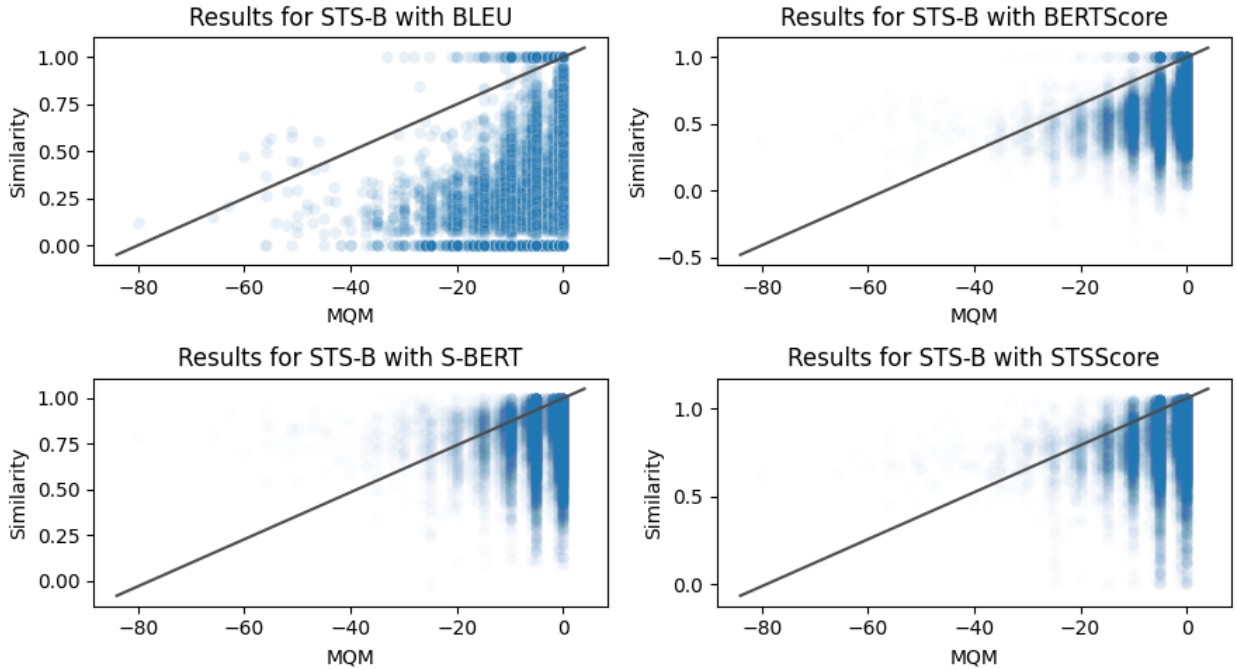

Figure 5: Evaluation of similarity measures on the MQM labelled test data for translations from Chinese to English (ZH-EN) WMT-22 metrics challenge. Ideally, the similarity correlates linearly with the MQM labels, i.e., the scores are close to the black line.

embedding approaches clearly outperform BLEU, which is again only weakly correlated with the expectation and has many similarity values of exactly zero.

## 4   Discussion

Our most important result is that our experiments support our hypothesis: for three data sets, the transformer-based prediction approach STSScorer aligned best with the expectation from the data, and we suggest using such approaches for future calculations of the semantic similarity. The exception is the translation task WMT-22, where STSScorer also aligned with our expectations, but where BERTScore overall was better aligned with the human MQM judgments. The weaker correlations on the WMT22-ZH-EN data show that considering only a single *generic* semantic similarity is not sufficient for more complex scenarios that consider different views on semantic similarity (e.g., locale style). For the STS-B, MRPC, and QQP data especially, the S-BERT approach also yields good results, though it seems to have a tendency to overestimate the similarity, even on the QQP data which was partially seen during training. BERTScore also yields promising results, but the scores are only moderately correlated with human labeled data (as measured with STS-B) and fail to fully capture semantic equality (as shown with MRPC and QQP). As could be expected, older n-gram based approaches like BLEU are not comparable and yield significantly worse results: the distribution of the similarity scores is overall fairly random, although the general trend of the labels can still be (weakly) observed. This means that rankings based on BLEU are likely not misleading, if a large amount of data is used and the effects between different models are large.

Another important aspect that we want to stress is that the consideration of the actual distributions of the similarities computed by the different models gave us insights well beyond the statistical markers. Just looking at the statistics, we would have missed important aspects like the tendency of BLEU towards values of exactly zero, the tendency of BERTScore to not yield very large values close to one for semantically equal statements, or the differences between STSScorer and S-BERT on paraphrases within the MRPC data.

We further note that our goal was to evaluate our hypothesis and provide an easy method for better semantic scoring. As a consequence, we choose to simply re-use an existing model from Huggingface instead of training our own model or using a checkpoint from elsewhere. While the model we use is not a lot worse than the top performers in the GLUE benchmark (best performing models in STS-B achieve correlations of up to 0.93), it cannot be considered as state-of-the-art for STS-B. Nevertheless, our results hold, i.e., STSScorer is a very good approach, though plugging in other models fine-tuned for semantic similarity prediction might yield (slightly) better results. We also note that Reimers & Gurevych (2019) also considered a fine-tuned version of S-BERT on the STS-B data by using the STS-B scores divided by five (same as us) to define a cosine similarity loss, which seemed to improve the alignment with STS-B when computing the cosine similarity. Nevertheless, based on the results reported by Reimers & Gurevych (2019), using the predictions directly should still be a better similarity measure than computing the cosine similarity of the embeddings. However, embeddings have the advantage, that they also enable other kinds of analysis, e.g., clustering or visualizations.

Another aspect to consider when computing the semantic similarity is the computational effort. The three embedding-based approaches are almost the same in terms of computational effort, as they are all based on the BERT architecture. As the authors of BERTScore note, the speed of is only slightly slower than that of BLEU (Zhang et al., 2020). Thus, the efficiency is not a major factor when choosing between these models.

## 5 Limitations

Even though we believe that STSScorer is currently the best method for computing similarities, there is also an important drawback: when we use transformer-based approaches to evaluate other – likely transformer-based – approaches, the evaluation will possibly have all the problems regarding biases that transformers have. Thus, we may further encode these biases, because we selected models based on a biased evaluator.

Additionally, all the current work ignores that semantic similarity is not absolute and also depends on the perspective of humans and can, e.g., be influenced by culture, social background, and other aspects, raising the question of whether we should rather specify our notion of similarity more carefully in the future and use multiple similarity measures that can capture such differences. This is also shown on the WMT22-ZH-EN task, where not only the accuracy influences the translation quality, but also other aspects like the use of locale conventions.

Another notable limitation is that semantic similarity data needs to be available to target languages, i.e., we cannot easily use a model that is fine-tuned on the English STS data to compute the semantic similarity in other languages, or even between languages. Approaches based on the cosine similarity of embeddings are more flexible in this regard, as they only require a suitable foundation model for the language, possibly augmented with further self-supervised pre-training as is done by S-BERT (Reimers & Gurevych, 2019).

## 6 Conclusion

The jumps in performance for natural language processing enable us to directly predict the semantic similarity instead of using embedding-based approaches or heuristics. Due to the readily available models on platforms like Huggingface, switching to such models for the future evaluation of the semantic similarity of results should be easily possible.

Future work should address the current limitations, most notably, the general problem capturing semantics with a single number – regardless of the approach! – is likely problematic. To resolve this, similar models to STSScore can be created that consider different aspects of semantics, e.g., for different aspects of translation quality based on MQM (Burchardt, 2013) or based on Leech's seven types of meaning (Leech, 1974). Any such approach should carefully consider potential biases and their effect on the created semantic similarity measures.

**Broader Impact Statement**

For many low-level tasks, we now have very good classification and regression models. We should consider using these models more often as performance measures for more complex tasks. We demonstrate this in this paper by showing that fine-tuned semantic similarity models are better at capturing expected characteristics of semantic similarity than relying only on the cosine-similarity of embeddings. However, when doing this, we risk that biases and limitations from the fine-tuned model affect the evaluation of complex task in an adversarial manner, which should also be further studied.

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

## A    Results of additional experiments

### A.1    Results when using BERT-base for STSScorer

In order to compare how much impact our choice of using a fine-tuned RoBERTa model has on the results, we provide a comparison with a BERT-base model instead. Table 2 shows the performance in comparison to RoBERTa. Figures 6 to 9 show the plots for the BERT-base model. The results indicate a clear gap, i.e., the better score on the STS-B data of the larger and better trained RoBERTa model robustly translates into a better semantic similarity estimation.

### A.2    Results when using an ensemble for scoring

We also provide results for using an ensemble of STSScore, BERTScore, and S-BERT. For this ensemble, we simply average over the similarity of the three embedding-based approaches. Table 3 shows the performance in comparison to the individual embedding based approaches. Figures 10 to 13 show the plots in comparison to the individual models. The ensemble is always between the two best models, i.e., it has some upside regarding robustness, but in general, it seems better to use STSScore directly.

### A.3    Impact of sequence length

Table 4 shows the impact of the sequence length measured in the number of characters on the STSScore. For this, we split the data into two (about) equally large subsets, such that the sequences shorter than the median are in one subset and the sequences equal to or longer than the median in the other. The median sequence lengths are 44 for STS-B, 116 for MRPC, 52 for QQP, and 123 for WMT22-ZH-EN. We do not observe a stable trend when considering the impact on the length. For WMT22-ZH-EN, the values on both subsets are larger, which indicates that this may be a case of the Simpson paradox.

|  | STS-B | | MRPC | | QQP | | WMT22-ZH-EN | |
|---|---|---|---|---|---|---|---|---|
|  | $r$ | $\rho$ | neg | pos | neg | pos | $r$ | $\rho$ |
| RoBERTa | 0.90 | 0.89 | 0.61 (0.18) | 0.84 (0.13) | 0.44 (0.24) | 0.76 (0.18) | 0.27 | 0.34 |
| BERT-base | 0.83 | 0.82 | 0.62 (0.18) | 0.79 (0.11) | 0.46 (0.23) | 0.69 (0.15) | 0.08 | 0.18 |

Table 2: Summary statistics of the results for the STSScore with different foundation models that were fine-tuned for the STS-B data. Pearson's $r$ and Spearman's $\rho$ between the labels and similarities for STS-B and WMT22-ZH-EN. Mean values with standard deviation in brackets for both classes of the MRPC and QQP data. We use *neg/pos* to indicate the classes such that *pos* is the semantically equal class. All values are rounded to the second digit.

|  | STS-B | | MRPC | | QQP | | WMT22-ZH-EN | |
|---|---|---|---|---|---|---|---|---|
|  | $r$ | $\rho$ | neg | pos | neg | pos | $r$ | $\rho$ |
| BERTScore | 0.53 | 0.53 | 0.53 (0.15) | 0.68 (0.13) | 0.44 (0.26) | 0.67 (0.17) | 0.32 | 0.37 |
| S-BERT | 0.83 | 0.82 | 0.71 (0.15) | 0.83 (0.12) | 0.56 (0.23) | 0.86 (0.10) | 0.19 | 0.24 |
| STSScore | 0.90 | 0.89 | 0.61 (0.18) | 0.84 (0.13) | 0.44 (0.24) | 0.76 (0.18) | 0.27 | 0.34 |
| Ensemble | 0.88 | 0.87 | 0.62 (0.13) | 0.78 (0.10) | 0.48 (0.24) | 0.77 (0.13) | 0.30 | 0.35 |

Table 3: Summary statistics of the results. Pearson's $r$ and Spearman's $\rho$ between the labels and similarities for STS-B and WMT22-ZH-EN. Mean values with standard deviation in brackets for both classes of the MRPC and QQP data. We use *neg/pos* to indicate the classes such that *pos* is the semantically equal class. All values are rounded to the second digit.

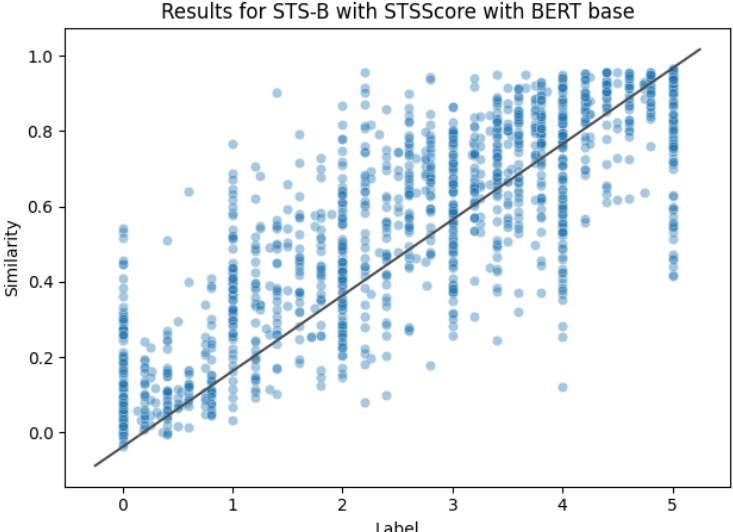

Figure 6: Evaluation of similarity measures on the test data of STS-B with BERT-base. Ideally, the similarity correlates linearly with the labels, i.e., the scores are close to the black line.

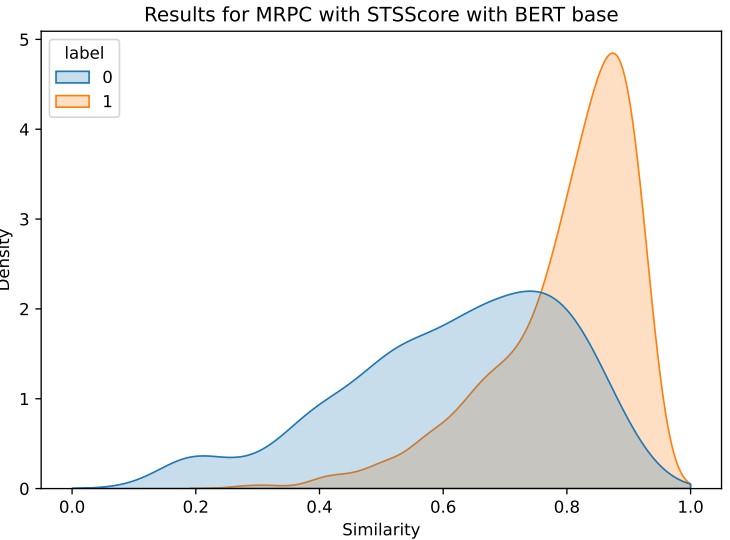

Figure 7: Evaluation of similarity measures on the test data of MRPC with BERT-base. Ideally, the positive class (1) has scores close to one and the negative class (0) has smaller values, but not close to zero.

| | STS-B | | MRPC | | QQP | | WMT22-ZH-EN | |
| | $r$ | $\rho$ | neg | pos | neg | pos | $r$ | $\rho$ |
|---|---|---|---|---|---|---|---|---|
| All | 0.90 | 0.89 | 0.61 (0.18) | 0.84 (0.13) | 0.44 (0.24) | 0.76 (0.18) | 0.27 | 0.34 |
| Shorter | 0.90 | 0.90 | 0.57 (0.18) | 0.80 (0.14) | 0.48 (0.24) | 0.78 (0.18) | 0.30 | 0.31 |
| Longer | 0.89 | 0.88 | 0.68 (0.15) | 0.86 (0.12) | 0.41 (0.23) | 0.74 (0.18) | 0.40 | 0.42 |

Table 4: Summary statistics of the results. Pearson's $r$ and Spearman's $\rho$ between the labels and similarities for STS-B and WMT22-ZH-EN. Mean values with standard deviation in brackets for both classes of the MRPC and QQP data. We use neg/pos to indicate the classes such that pos is the semantically equal class. All values are rounded to the second digit.

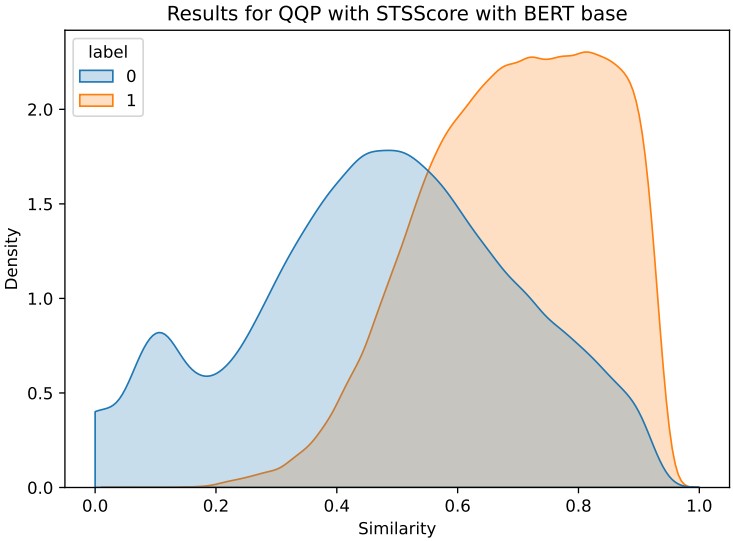

Figure 8: Evaluation of similarity measures on the training data of QQP with BERT-base. Ideally, the positive class (1) has scores close to one and the negative class (0) has smaller values, with only a small fraction being close to zero.

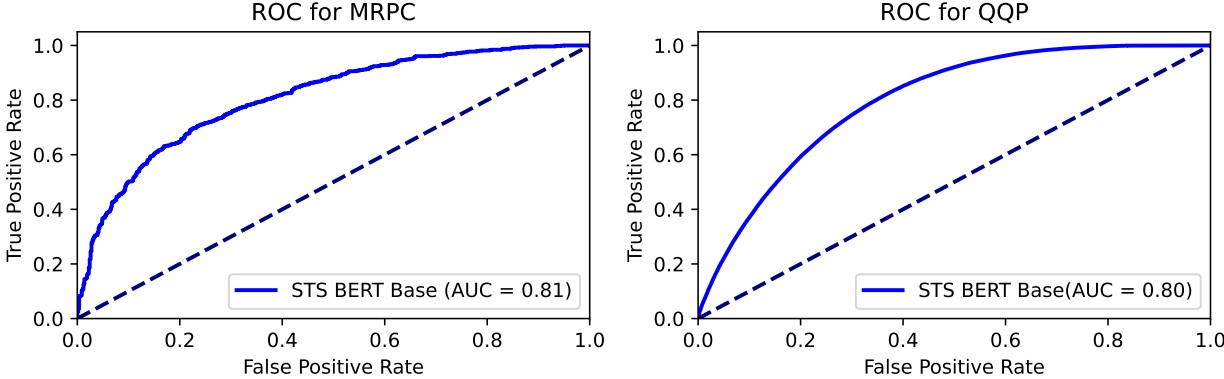

Figure 9: Receiver Operator Characteristics (ROC) and AUC measurements for using the semantic similarity metrics as classifiers for the MRPC and QQP data with BERT-base. A larger area is better.

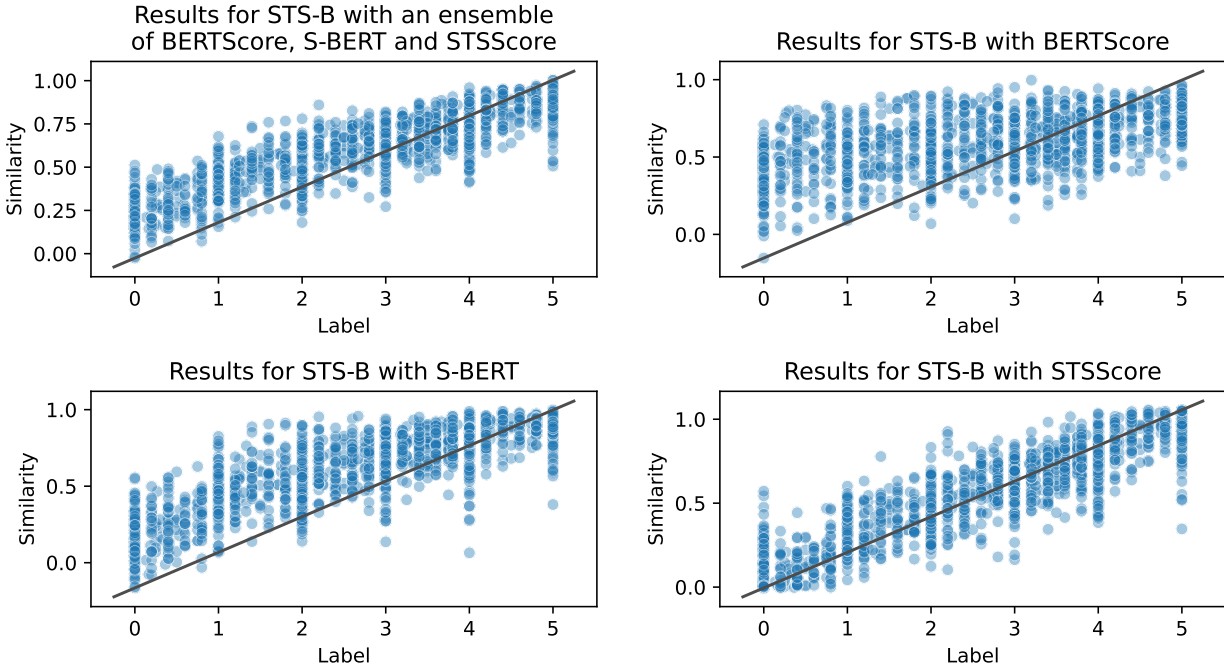

Figure 10: Evaluation of similarity measures on the test data of STS-B with an ensemble of the embedding based approaches. Ideally, the similarity correlates linearly with the labels, i.e., the scores are close to the black line.

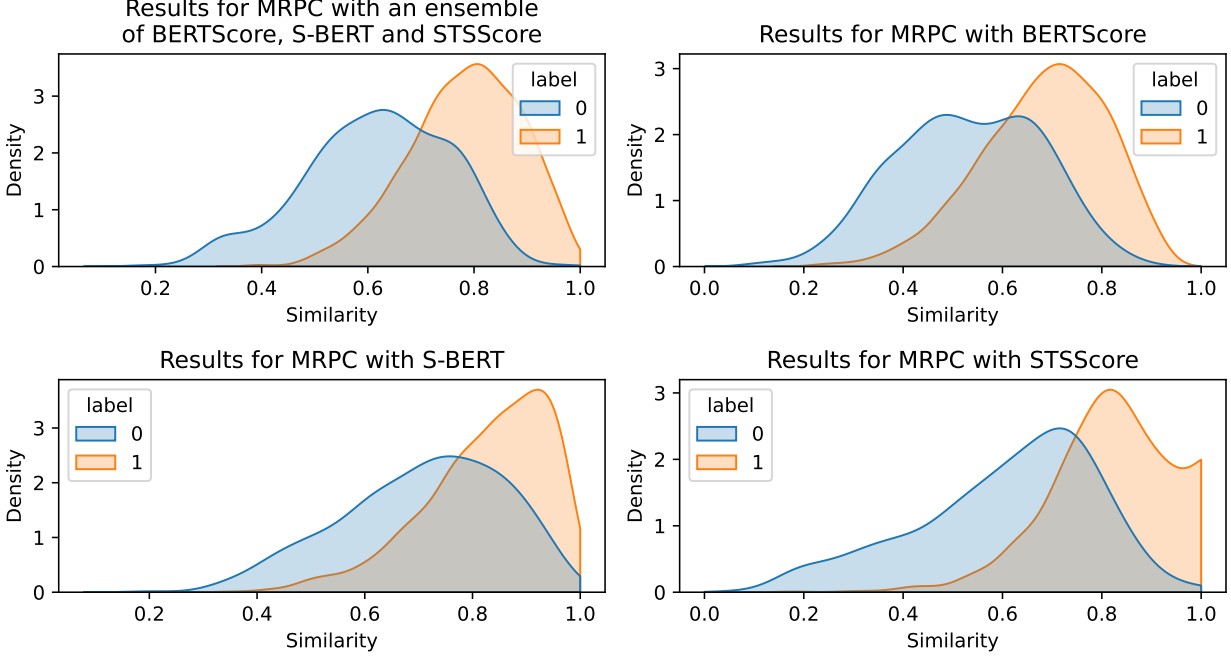

Figure 11: Evaluation of similarity measures on the test data of MRPC with an ensemble of the embedding based approaches. Ideally, the positive class (1) has scores close to one and the negative class (0) has smaller values, but not close to zero.

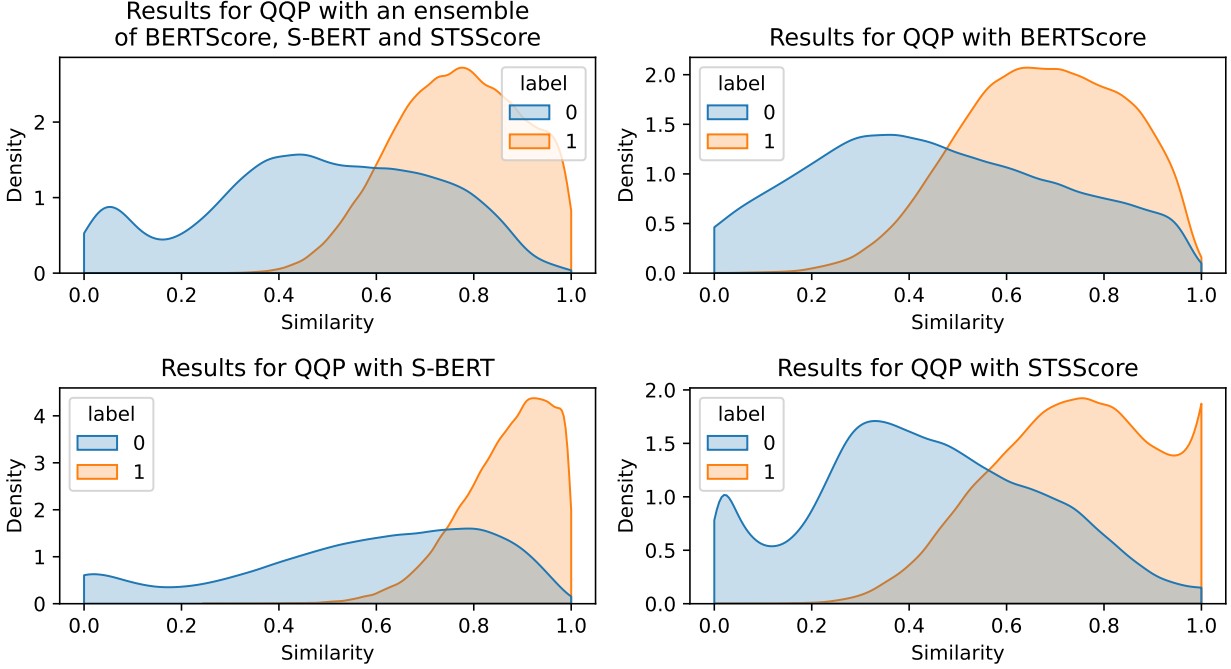

Figure 12: Evaluation of similarity measures on the training data of QQP with an ensemble of the embedding based approaches. Ideally, the positive class (1) has scores close to one and the negative class (0) has smaller values, with only a small fraction being close to zero.

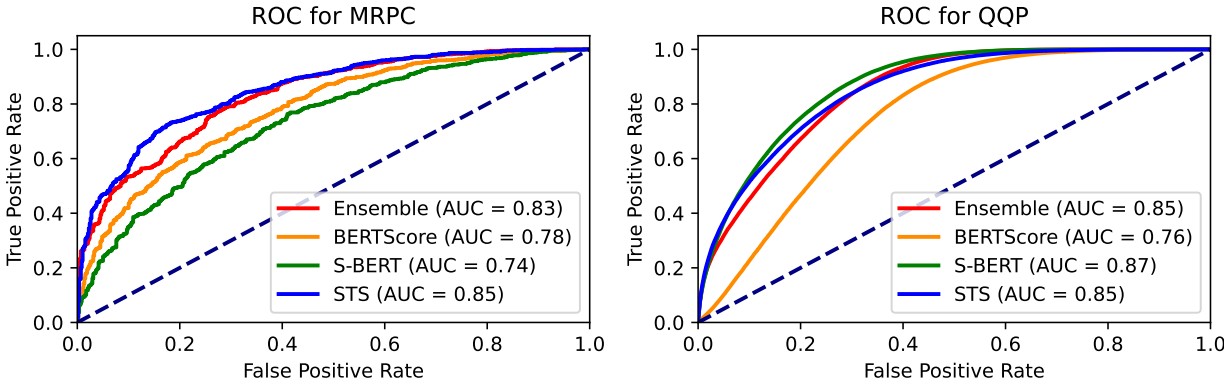

Figure 13: Receiver Operator Characteristics (ROC) and AUC measurements for using the semantic similarity metrics as classifiers for the MRPC and QQP data with an ensemble of the embedding based approaches. A larger area is better.

