# OpenReview forum: "Semantic similarity prediction is better than other semantic similarity measures"
_TMLR — Accepted by TMLR_

### Review · Reviewer_2kRv · 2023-11-19

**Summary Of Contributions:**

This paper evaluates the use of BERT-based models fine-tuned for regression on predicting semantic similarity. The paper investigates how well this approach can serve as a metric for similarity for tasks like paraphrasing and compares against other common approaches like BLEU and BERTScore. The paper finds that the regression-based approaches outperform existing metrics on a number of these tasks.

**Audience:**

Yes

**Claims And Evidence:**

Yes

**Requested Changes:**

Major changes:
* Especially because the approach itself is not proposed by the paper, there should be much more thorough analysis of the metric beyond what is provided in the paper. E.g., evaluating on the myriad other tasks that rely on measures of similarity, like machine translation or image captioning.

Medium changes:
* STSScorer should at the very least be described in more detail in terms of how it was trained, even if the authors didn't train it. E.g., it is finetuned with a regression objective, right?

Minor changes:
* "STS-B" should be introduced before using the acronym / referring to it, especially in the abstract.
* Typo: "BiLinigual Evaluation Understudy" --> "Bilingual"
* Typo: "since its publications in 2020" --> "since its publication in 2020"

**Strengths And Weaknesses:**

Strengths:
- The experimental setup is quite thoroughly described, including definitions of the hypotheses and potential confoundres.
- Examining the predicted distributions of similarity is interesting.

Weaknesses:
- The contribution is narrow. The model evaluated is already trained for the task it is evaluated on, and just downloaded from HF.
- Semantic similarity is notoriously subjective (Goldstone et al. 1997, Similarity in context). This is a bigger problem with the framing of semantic similarity evaluation that goes beyond this particular paper, as discussed at the end of section 5. But more discussion throughout would be nice. E.g., it's unclear what labels the pairs of sentences in 3.0.2 _should_ have, because it's so relative.
- The evaluation is quite narrow scope, just on three datasets which are essentially for paraphrase detection. Meanwhile, BERTScore also evaluates on tasks like machine translation and image captioning. Why not evaluate the proposed metric on more downstream tasks?

---

> ### Author Response · Authors · 2023-12-04
> **Response to review**
>
> Thanks for your review. We tried to address everything and respond to the requested changes below. To further help with checking our response, the supplemental material contains a PDF, in which all changes (except typo corrections) for this revision are marked in blue.
>
> Request 1: Major changes: Especially because the approach itself is not proposed by the paper, there should be much more thorough analysis of the metric beyond what is provided in the paper. E.g., evaluating on the myriad other tasks that rely on measures of similarity, like machine translation or image captioning.
>
> Answer 1: We extended our analysis with machine translation data (WMT22 shared metrics task, ZH-EN, MQM labels, see Reviewer KAVy, comment 1). While adding further data sets would be nice, the type of in-depth analysis we conduct means that we run out of space fast. We know that this is a weak reason, but we want to rather report on the data we consider in detail, than superficially on a larger data set, because we believe that this is one aspect that sets this paper apart from many other analyses. Notably, the direct consideration of the distribution adds a lot of value, as we can observe some interesting aspects like BERTScores tendency to not yield values close to 1.0.
>
> Request 2: Medium changes: STSScorer should at the very least be described in more detail in terms of how it was trained, even if the authors didn't train it. E.g., it is finetuned with a regression objective, right?
>
> Answer 2: We extended the section on the tools we used with the missing details: “The RoBERTa model we used for STSScore (Held, 2020) was trained with a learning rate of $2\cdot 10^{-5}$ with a linear scheduler and a warmup ratio of 0.06 for 10 epochs using an Adam optimizer with $\beta_1=0.9,\beta_2=0.999$ and $\epsilon=10^{-8}$ using MSE loss.”
>
> Request 3: Minor changes:
> "STS-B" should be introduced before using the acronym / referring to it, especially in the abstract.
> Typo: "BiLinigual Evaluation Understudy" --> "Bilingual"
> Typo: "since its publications in 2020" --> "since its publication in 2020"
>
> Answer 3: We introduce the acronym in the abstract and fixed the second typo. However, the uppercase “L” in BiLingual is intentional, as this is the BL part of the acronym BLEU.

---

> > ### Comment · Reviewer_2kRv · 2023-12-07
> >
> > > While adding further data sets would be nice, the type of in-depth analysis we conduct means that we run out of space fast. We know that this is a weak reason, but we want to rather report on the data we consider in detail, than superficially on a larger data set, because we believe that this is one aspect that sets this paper apart from many other analyses.
> >
> > There is always a trade-off between depth and breadth in analysis and evaluation. I agree the in-depth analysis provided by this paper is valuable. But if this metric is being proposed as one better than existing similarity metrics, the evaluation also needs to be comprehensive (i.e. evaluated across numerous datasets like the other papers have evaluated on). In my opinion, the paper is currently not very dense, and there is significant room for adding more material within the page limits.
> >
> > > We extended the section on the tools we used with the missing details
> >
> > I meant here specifically what the objective is (which includes the format of the data being used). If it is a regression objective, from where are the labels derived? What do training data points look like? A function/equation showing how loss is computed for a single example would answer the question I originally posed.
> >
> > > "BiLingual"
> >
> > The typo was specifically that there was an extra "i" as in "BiLinigual"

---

> > > ### Author Response · Authors · 2023-12-08
> > >
> > > Sorry, for misunderstanding about the objective. In addition the the hyperparameters that implicitly specify this for the concrete model we used, we now further extended Section 2.1, which already describes the data, as follows (bold new):
> > >
> > > "The STS-B task contains sentence pairs from news, image captions and Web forums. Each sentence pair received a score between zero and five. These scores were computed as the average of the semantic similarity rating conducted by three humans such that five means that the raters believe the sentences mean exactly the same and zero means that the sentences are completely unrelated to each other. We simply use a **regression** model trained for this task and divide the results by five to scale them to the interval $[0,1]$. **The regression model is trained with Mean Squared Error (MSE) loss, which is defined as** $MSE(y,y^*)=(y-y^*)^2$, **where** $y$ **is the predicted label and** $y^*$ **the expected label.** Hereafter, we refer to this model as STSScorer."
> > >
> > > The BiLingual typo is now also fixed, we missed the additional I and misinterpreted that comment.

---

### Review · Reviewer_KAVy · 2023-11-21

**Summary Of Contributions:**

This paper addresses the problem of **measuring semantic similarity**: Given two natural language texts A and B, we would like to measure how similar A and B are in terms of their meaning (irrespective of any superficial similarities or differences in the lexical surface form). Given the centrality of this problem in NLP --- particularly for evaluating how "good" / similar a model's output is with respect to a human-generated reference output --- various metrics have been proposed to address this problem throughout the years. Whereas earlier solutions have mostly focused on **lexical** similarity and overlap (e.g. BLEU, ROUGE, etc.), recent work has demonstrated the superiority of **contextual embedding-based similarity metrics**, such as sentence-BERT. Unlike metrics that are based on surface lexical overlap and similarity, these similarity-based metrics are instead computed by contextually encoding each text with a Transformer model, and measuring the similarity between the two contextual encoding of each text in the dense vector space.

To that end, this paper proposes a new, simple way of measuring semantic similarity. Rather than directly measuring the contextual embedding similarity in the dense vector space with an off-the-shelf pre-trained model like BERT or RoBERTa directly, the paper proposes to instead use the resulting similarity metric from a **RoBERTa model that is fine-tuned on the STS-B task**. As the STS-B task: (i) is designed to train the model to do well on semantic similarity prediction, and (ii) requires the model to output a semantic similarity score that ranges from 1 to 5, the paper argues that the output ratings from a fine-tuned RoBERTa model on STS-B (denoted as **STSScorer**) provides a natural, simple, and off-the-shelf way for measuring textual similarity, even when it is applied to other, more general text domains beyond the STS-B dataset. By directly leveraging the STS-B semantic similarity rating scale, this method circumvents the need for choosing additional vector similarity metrics, such as cosine similarity.

Experiments on STS-B, MRPC, and QQP suggest that the resulting similarity score distributions from the STSScorer is better or equally correlated with human judgment compared to other alternative metrics and baselines, such as BERTScore and Sentence-BERT.

**Audience:**

Yes

**Broader Impact Concerns:**

No broader impact concerns come to mind.

**Claims And Evidence:**

No

**Requested Changes:**

1. **Critical**: Evaluate the metric on a much broader range of tasks, including for evaluating machine translation outputs vs the gold reference, and also for text summarisation, etc. This will empirically validate whether or not the proposed metric is truly a general-purpose metric for conducting semantic similarity on text, rather than something that only works well for text domains and NLP tasks that are similar to STS-B. It would also be nice to compare against existing contextual-embedding-vector-based metric for these tasks,for which prior work has dedicated a lot of work in the past (e.g. the WMT Shared Task on Quality Estimation).

2. **Critical**: Report performance on tasks with binary labels (e.g. MRPC, QQP), where we need to discretise the predictions (e.g. an STSScore of >0.7 means the paraphrase label is positive, <0.7 means negative, etc.). Also come up with a scheme that would enable us to set the appropriate threshold for such prediction on a dataset-by-dataset basis, because arguably the appropriate threshold will be different for each dataset.

3. **Critical**: Add additional analyses and ablation studies, such as how good the similarity metric is for long vs short sentences, whether or not we can ensemble & combine STSScore with other metrics to get a better distribution that more closely reflects the human similarity rating distribution, etc.

4. **Recommended**: Incorporating the suggestions and resolving the questions that I raised above, and also careful copy-editing of the paper.

**Strengths And Weaknesses:**

**Strengths**

1. The problem of measuring semantic similarity is an important one in the field: Better semantic similarity metrics would enable us to more accurately evaluate our current models, truly understand the extent of our progress, and also enable us to design better retrieval systems and their various applications (e.g. which of the documents that we have are the most / least similar to this particular document that we care about?).

2. The approach is very simple and easy to implement, and leverages an off-the-shelf pre-trained RoBERTa model that is fine-tuned on STS-B, which is readily available for the general public on libraries like HuggingFace.

3. The research hypothesis is clearly stated on page 2, which helps crystallise what the claim is and what is being done.

**Weaknesses**

1. The main weakness of this paper is that the empirical scope of the paper is very narrow, and more concerningly, does not validate the claim that is being made. Concretely, the claim is that the semantic similarity scores from STS-B can be **broadly and generally** applied to evaluate the semantic similarity of two natural language texts, even in other domains that are quite different from the STS-B dataset. However, the paper only evaluates on the STS-B test set (which is in-domain), in addition to MRPC and QQP. The problem is that the MRPC and QQP datasets are: (i) both semantic similarity prediction tasks (which may not be representative of other types of NLP tasks), and (ii) only cover newswire text (MRPC) and web text (QQP). To what extent would the proposed semantic similarity metric work well for evaluating, e.g. machine translation outputs, or text summarisation outputs against the reference, human-generated output? And how would this metric compare with BLEU (default evaluation metric for machine translation) and ROUGE (default metric for text summarisation) for those tasks? And can the metric work well when being used to evaluate other domains / more noisy natural language text, such as Reddit / Twitter? Given the small size of the STS-B training set, I am skeptical that RoBERTa fine-tuned on STS-B would necessarily provide better evaluation metrics for other important NLP tasks, use cases, and domains (unless otherwise empirically shown, which the paper in its current form has not done yet by virtue of evaluating only on 3 mostly-similar semantic similarity prediction tasks that are not representative of many different NLU & NLG system use cases).

2. For the case where the metric has to be used to provide negative or positive labels (e.g. given text "A" and "B", output a label of 1 if they are semantically paraphrases, or "0" otherwise), it is still very much unclear how we can **set the appropriate threshold**, which might differ from dataset to dataset. For instance, for MRPC, an STSScore of 0.7 and above should mean a label of 1 (paraphrase), whereas an STSScore of <0.7 should mean a label of 0 (not paraphrase). Can we come up with such a decision rule using this metric? If so, how should we determine this threshold, which would likely be dataset-specific? And how well do we perform on the binary classification task after setting that threshold (i.e. what MRPC accuracy do we get when using the threshold to determine whether or not two texts are paraphrases of one another?).

3. Given the simplicity of the approach (which is a good thing as simple techniques can be adopted more easily), I would expect there to be more substantive content & analyses given the fairly generous 10-page content, such as: (i) experiments on more tasks (including on evaluating generation-based tasks like machine translation, summarisation, etc. which would be very useful to validate the claimed generality of the approach), (ii) ablation studies (how does the similarity metric performance differ for long vs short text, etc.), (iii) experiments with combining multiple metrics (e.g. can we combine STSScore and S-BERT to yield a better semantic similarity metric than either one of them alone), etc. In this paper, 2 full pages (pages 6 & 9) are dedicated to plots that could be much smaller and stacked together; this would then free up space for more analyses, etc. Also, a lot of the space is dedicated to explaining existing tasks (MRPC & QQP) in great detail, which is not central to the paper's contributions and could be described more succinctly in the main paper, with the details deferred to the Appendix.

4.  For Figures 2 & 3, it would be nice to see a quantification of the "overlap area" between the negative and positive labels, with the assumption that the less overlap -> the better (i.e. the distributions of similarity for positive and negative classes are quite distinct from one another). This would allow us to better quantify the superiority of the STSScore compared to the baseline metrics.

5. The paper has a non-trivial amount of typos and grammatical mistakes, and could benefit from careful copy-editing.

**Questions and Suggestions**

1. On page 1, it is stated that "The transformer architecture enabled the context-sensitive calculation of embeddings". This is incorrect: One can compute context-sensitive calculation of embeddings from other models like LSTMs or bidirectional LSTMs, for instance, and still have pretty good results. See the contextual word embedding of the ELMo model of Peters et al. (2018), which was originally based on bidirectional LSTMs. So the ability to do context-sensitive calculation of embeddings is not at all specific to Transformers. However, the Transformer architecture enables us to **do this at scale** by using a larger model size and pre-training on a larger dataset (e.g. compared to LSTMs), because we can parallelise across the time dimension and make better use of specialised hardwares like GPUs.

2.  Some of the figures take up way too much space (e.g. Figures 2 and 3). It would be better to make them smaller and combine them to make room for other analyses, etc.

**Typos / Presentation Suggestions**

1. In page 1, "ROGUE" should be "ROUGE".
2. In page 4, "were created independent ..." should be "were created **independently** ".
3. In page 4, "As example, consider ..." should be "For example" or "As an example" .
4. In page 8, "The questions indicates" -> "indicate".
5. In page 8, "is again exhibit" is a grammatical error.
6. In page 10, "a lot worse then" -> "than".

---

> ### Author Response · Authors · 2023-11-27
>
> Hi! Thanks for taking the time to review our work. We are currently planning the revision. If you have time, we have one question regarding the requested changes for the WMT task.
>
> As far as we know, the quality estimation tasks do not provide human judgments that compare a solution to a reference, due to which we are not sure what to use a ground truth. Instead, we could use the WMT shared metrics task data from 2022, with all translations to English. Here, we have human annotations regarding the quality of a translation with respect to a reference translation. We could then use pairs of the references and alternative translations, to determine the correlation with the human judgements.
>
> We then just need to consider how exactly to interpret these human judgements, because as far we understood them they are not rated on some fixed scale, but are rather a sum of errors, where different kinds of errors have different impacts.

---

> ### Author Response · Authors · 2023-12-04
> **Response to review**
>
> Thanks for your review. We tried to address everything and respond to the requested changes below. To further help with checking our response, the supplemental material contains a PDF, in which all changes (except typo corrections) for this revision are marked in blue.
>
> Request 1: Critical: Evaluate the metric on a much broader range of tasks, including for evaluating machine translation outputs vs the gold reference, and also for text summarisation, etc. This will empirically validate whether or not the proposed metric is truly a general-purpose metric for conducting semantic similarity on text, rather than something that only works well for text domains and NLP tasks that are similar to STS-B. It would also be nice to compare against existing contextual-embedding-vector-based metric for these tasks,for which prior work has dedicated a lot of work in the past (e.g. the WMT Shared Task on Quality Estimation).
>
> Answer 1: We have added data from the WMT22 metrics task as a fourth data set. We could only use the data for ZH-EN because this was the only English data from that year with MQM labels. We formulated our expectations on this data based on the assumption that the MQM does not only measure the semantic similarity (accuracy) but also other aspects related to translation quality (e.g., fluency, locale) and expect that the correlations will be weaker than for the other tasks, but visible nevertheless. This is also roughly what we observed. We were a bit surprised that BERTScore was the best model for this task, but this can possibly be explained by how MQM works.
>
> Request 2: Critical: Report performance on tasks with binary labels (e.g. MRPC, QQP), where we need to discretise the predictions (e.g. an STSScore of >0.7 means the paraphrase label is positive, <0.7 means negative, etc.). Also come up with a scheme that would enable us to set the appropriate threshold for such prediction on a dataset-by-dataset basis, because arguably the appropriate threshold will be different for each dataset.
>
> Answer 2: Looking at the performance for binary predictions of the scores is a great idea, as this operationalizes the meaningful part of the overlap between the distributions for the different classes. However, instead of determining a threshold, we decided to evaluate this in a threshold-independent manner by adding ROC curves and computing the AUC. We have extended the analysis in the paper accordingly and the results are completely in line with the analysis of the distributions of the scores.
>
> Request 3: Critical: Add additional analyses and ablation studies, such as how good the similarity metric is for long vs short sentences, whether or not we can ensemble & combine STSScore with other metrics to get a better distribution that more closely reflects the human similarity rating distribution, etc.
>
> Answer 3: We extended our online materials with the requested ablation studies. In short, the sequence length seems to have, for some data sets, an impact on the similarities (we split by the median), but these differences are consistent with our results, e.g., for MRPC, the absolute values are a bit higher for longer sequences, but the difference between the classes is conserved.
> The ensembles are fairly good, but always perform worse than STSScorer, except for the translation task where they have a (very small) advantage. This is to be expected, given that for most tasks, the weaker BERTScore affects the results negatively, except for the translation tasks, where it has a positive impact.
>
> Request 4: Recommended: Incorporating the suggestions and resolving the questions that I raised above, and also careful copy-editing of the paper.
>
> Answer 4: Thank you for your valuable suggestions. These have strengthened the paper considerably. We also conducted another round of copy-editing.

---

> > ### Comment · Reviewer_KAVy · 2023-12-13
> > **Thank You for the Response & Revision**
> >
> > Thank you for the authors' response and the revision. I appreciate that the revised version has now included results from the WMT22 Metrics Challenge. The result improves my confidence that the metric is more general than initially shown.
> >
> > I have a few follow-up questions:
> >
> > 1. The ensemble experiment (i.e. combining multiple metrics together to better assess different, complementary aspects of semantic similarity) is mentioned, but I don't see the results in the revised version. Did ensembling STSScore with other metrics yield better results?
> >
> > 2. Another reviewer raised the issue of using the metric for other languages, which is not currently feasible due to the lack of STS-B data in other languages. This seems to affect the generality of the approach; in contrast, other BERT-based metrics that are based on embedding similarity can be applied to other languages (assuming the new languages are covered by multi-lingual BERT). Do you have ideas for how to overcome this limitation?

---

> > > ### Author Response · Authors · 2023-12-13
> > >
> > > 1) The results for the ensemble are in the supplemental material in the eval-semantic-similarity.ipynb. We decided against adding this in the appendix, to keep the PDF clean, but we could easily add the results figures. The ensembles are fairly good, but always perform worse than STSScorer, except for the translation task where they have a (very small) advantage. This is to be expected, given that for most tasks, the weaker BERTScore affects the results negatively, except for the translation tasks, where it has a positive impact.
> > >
> > > 2) This is indeed a large limitation for the use of this approach beyond English. This is added as limitation in Section 5. The only real way is to have semantic similarity data for other languages as well. Notably, such data should be created anyways before using any approach for any language anyways, to evaluate the semantic similarity measure. So while we cannot use such an approach in another language easily, we should use other approaches with care anyways.

---

### Review · Reviewer_aEt1 · 2023-11-23

**Summary Of Contributions:**

The paper presents a novel method for predicting semantic similarity between natural language texts called STSScore, leveraging a fine-tuned model for the task from the GLUE benchmark. The authors discuss existing methods for measuring semantic similarity, such as BLEU and methods based on various embeddings like BERTScore and S-BERT, and argue for a regression model instead. The proposed model, STSScore, is built upon a fine-tuned RoBERTa model for the STS-B task. The authors conducted a confirmatory study to test their hypothesis that their approach, based on predicting the semantic similarity directly, is better aligned with expectations of a robust semantic similarity measure. In their analyses, they compared results from BLEU, BERTScore, S-BERT, and STSScore to evaluate the relationship between different measures across various tasks such as STS-B, Microsoft Research Paraphrase Corpus (MRPC), and Quora Question Pairs (QQP) data. The findings mostly aligned with their hypothesis, showcasing the potential of STSScore for semantic similarity tasks, although a detailed comparison against other potential models was not extensively discussed.

**Audience:**

Yes

**Claims And Evidence:**

Yes

**Requested Changes:**

1. Comparisons: Can you include comparisons using models other than RoBERTa if that's feasible?

2. Multilingual Analysis: I'd recommend the authors to extend their approach to languages other than English to test the robustness and versatility of STSScore.

3. Discuss Limitations: The paper has a distinct lack of discussion about limitations. Every approach has its limitations, and discussing them keeps the research balanced and provides a fuller picture of the method to the readers.

4. Computation Efficiency: Understanding the computation cost of the proposed approach in comparison to others is important. Please include a discussion or a benchmark of the computational efficiency of STSScore.

5. Future Work: Indicate directions for future research, such as how to improve the existing model or if there are any alternative approaches you would like to explore in the future.

**Strengths And Weaknesses:**

## Strengths
- The paper is well-written, effectively beginning by covering the motivation behind using a fine-tuned model for predicting semantic similarity.
- A comprehensive discussion is presented of the current methods for determining semantic similarity, which makes it clear why a new approach would be beneficial.
- The hypothesis---that using transformers for measuring semantic similarity might outperform other methods---is clearly stated and well-motivated.
- The authors provide a thorough approach to test their hypothesis, including calculations, visualizations, and detailed explanations of their analysis.
- The method, STSScore, itself appears promising, as it's based on a robust fine-tuned RoBERTa model.

## Weaknesses:
- The use of only one model (RoBERTa) without comparing the performance of other transformer architectures may lead to biased conclusions. Other models could have potentially affected results.
- The paper focuses only on the English language. It would be interesting to see a cross-linguistic analysis of the proposed method.
- There is very little discussion about the limitations of the proposed method

---

> ### Author Response · Authors · 2023-12-04
> **Response to reviews**
>
> Thanks for your review. We tried to address everything and respond to the requested changes below. To further help with checking our response, the supplemental material contains a PDF, in which all changes (except typo corrections) for this revision are marked in blue.
>
> Request 1: Comparisons: Can you include comparisons using models other than RoBERTa if that's feasible?
> Answer 1: Within the core paper, this would consume too much space. However, we have added a second Jupyter notebook to our online materials, that replicates the complete work with a BERT-base model as a comparison. The short summary is this: the results are similar, but a bit weaker than with RoBERTa, as can be expected based on the general performance difference between these two models.
>
> Request 2: Multilingual Analysis: I'd recommend the authors to extend their approach to languages other than English to test the robustness and versatility of STSScore.
> Answer 2: Unfortunately, we could not implement this request. For this, we would need two things: a suitable labeled set of data for semantic similarity in that language, like the STS data for English. Additionally, we would also need suitable benchmark sets like MRPC and QQP. We acknowledge this in the new limitations section (see below).
>
> Request 3: Discuss Limitations: The paper has a distinct lack of discussion about limitations. Every approach has its limitations, and discussing them keeps the research balanced and provides a fuller picture of the method to the readers.
> Answer 3: We added a new section with limitations after the discussion.
>
> Request 4: Computation Efficiency: Understanding the computation cost of the proposed approach in comparison to others is important. Please include a discussion or a benchmark of the computational efficiency of STSScore.
> Answer 4: We added the following to the discussion: “Another aspect to consider when computing the semantic similarity is the computational effort. The three embedding based approaches are almost the same in terms of computational effort, as they are all based on the BERT architecture. As the authors of BERTScore note, the speed of this approach is slightly slower than that of BLEU. Thus, the efficiency is not a major factor when choosing between these models.”
>
> Request 5: Future Work: Indicate directions for future research, such as how to improve the existing model or if there are any alternative approaches you would like to explore in the future.
> Answer 5: We added the following to the conclusion at the end of the paper:
> Future work should address the current limitations, most notably, the general problem of capturing semantics with a single number -- regardless of the approach! -- is likely problematic. To resolve this, similar models to STSScore can be created that consider different aspects of semantics, e.g., for different aspects of translation quality based on MQM (Burchardt, 2013) or based on Leech's seven types of meaning (Leech, 1974). Any such approach should carefully consider potential biases and their effect on the created semantic similarity measures.

---

> > ### Comment · Reviewer_aEt1 · 2023-12-19
> > **Thanks for the thoughtful response**
> >
> > Appreciate your response. I'm good with the edits you've made.

---

### Author Response · Authors · 2023-12-04
**Response to Reviews**

Thank you to all three reviewers. We tried to address all requested changes within a revision and prepared everything in a way that we believe facilitates a fast potential second round of reviews, before submitting a recommendation to the AE.

---

### Author Response · Authors · 2024-01-17

Thanks to the editor and all reviewers for the nice and smooth review process with great and helpful comments that improved the manuscript! The CR is now uploaded with the requested changes.

---

### Decision · Action_Editor_LMXn · 2024-01-15

**Recommendation:** Accept with minor revision

**Comment:**

This work studies the effectiveness of semantic similarity prediction as a more general metric. It is overall a rather narrow contribution, but it documents valuable evidence that may be useful for some of the community. The revision has also substantially improved the presentation. This paper thus clears the TMLR criteria and I recommend acceptance after a few minor revisions:

1) please fix the typo in the "Hypothesis" block which reads "capable oh robustly measuring"
2) please include a brief appendix in the PDF documenting the new results mentioned in the responses ("our online materials"). This is important for completeness and archival purposes.

**Audience:**

The intended audience is narrow, but it exists.

**Claims And Evidence:**

Reviewers overall find that the scope of this work is very narrow, and extending it (eg to other languages) is not straightforward. Nevertheless, after the revisions, the scope of the claims and the limitations are made clear. Therefore, we are in agreement that the claims made are well supported.